# Cepharanthine Dry Powder Inhaler for the Treatment of Acute Lung Injury

**DOI:** 10.3390/molecules28114441

**Published:** 2023-05-30

**Authors:** Di Liang, Wanmei Wang, Guangrui Chen, Jian Li, Guifang Dou, Hui Gan, Peng Han, Lina Du, Ruolan Gu

**Affiliations:** Department of Pharmaceutical Sciences, Beijing Institute of Radiation Medicine, Beijing 100850, China; liangdi0807@163.com (D.L.); wangwanmei20@sina.com (W.W.); chen_guangrui@163.com (G.C.); lijianjky@163.com (J.L.); dougf@bmi.ac.cn (G.D.); ganh2003@163.com (H.G.); 15901135949@163.com (P.H.)

**Keywords:** cepharanthine, pulmonary drug delivery, acute lung injury, dry powder inhalers

## Abstract

Severe acute respiratory syndrome Coronavirus 2 (SARS-CoV-2) induces a severe cytokine storm that may cause acute lung injury/acute respiratory distress syndrome (ALI/ARDS) with high clinical morbidity and mortality in infected individuals. Cepharanthine (CEP) is a bisbenzylisoquinoline alkaloid isolated and extracted from *Stephania cepharantha Hayata*. It exhibits various pharmacological effects, including antioxidant, anti-inflammatory, immunomodulatory, anti-tumor, and antiviral activities. The low oral bioavailability of CEP can be attributed to its poor water solubility. In this study, we utilized the freeze-drying method to prepare dry powder inhalers (DPI) for the treatment of acute lung injury (ALI) in rats via pulmonary administration. According to the powder properties study, the aerodynamic median diameter (D_a_) of the DPIs was 3.2 μm, and the in vitro lung deposition rate was 30.26; thus, meeting the Chinese Pharmacopoeia standard for pulmonary inhalation administration. We established an ALI rat model by intratracheal injection of hydrochloric acid (1.2 mL/kg, pH = 1.25). At 1 h after the model’s establishment, CEP dry powder inhalers (CEP DPIs) (30 mg/kg) were sprayed into the lungs of rats with ALI via the trachea. Compared with the model group, the treatment group exhibited a reduced pulmonary edema and hemorrhage, and significantly reduced content of inflammatory factors (TNF-α, IL-6 and total protein) in their lungs (*p* < 0.01), indicating that the main mechanism of CEP underlying the treatment of ALI is anti-inflammation. Overall, the dry powder inhaler can deliver the drug directly to the site of the disease, increasing the intrapulmonary utilization of CEP and improving its efficacy, making it a promising inhalable formulation for the treatment of ALI.

## 1. Introduction

Acute lung injury (ALI) is a lung disease characterized by a diffuse alveolar injury that results in an excessive inflammatory response and apoptosis of alveolar epithelial cells in the lungs [1]. Without timely treatment, it can progress to acute respiratory distress syndrome (ARDS) associated with increased mortality rates [2]. The most common causes of ALI are pneumonia and indirect ARDS sepsis [3]. Although most patients recover with normal or near-normal pulmonary physiology, they may exhibit significant and persistent muscle weakness, polyneuropathy, tracheal stenosis, contractures, a higher incidence of depression, anxiety, and posttraumatic stress disorder and significant cognitive impairment in executive function, memory, attention, and social isolation [4,5]. In December 2019, the coronavirus disease 2019 (COVID-19) began spreading worldwide. Its most notable pathological changes are observed in the lungs, leading to clinical symptoms such as ALI and ARDS [6]. Currently, there are no specific drugs approved for the clinical treatment of ALI. Nebulized inhaled hormonal and anti-inflammatory drugs are commonly used to alleviate ALI symptoms; however, their long-term use can produce adverse effects and tolerance [7,8]. Therefore, the prevention and treatment of ALI is an urgent problem that needs to be addressed.

Cepharanthine (CEP) (Figure 1) is a bisbenzylisoquinoline alkaloid mainly extracted from *Stephania cepharantha* Hayate [9]. In the chemical structure of CEP, there are two coclurine units connected head-to-head, giving it an elliptical macrocyclic structure. This lends it unique chemical properties such as ether solubility, optical activity, and the capacity to reduce the mobility of various biofilms [10,11]. It exhibits various physiological activities, such as anti-inflammatory, analgesic, and antiviral effects [10] and is widely used in the treatment of various acute and chronic diseases, including radiation-induced leukopenia, idiopathic thrombocytopenic purpura, pemphigus, furunculosis, venomous snake bites, dry mouth, nodular disease, refractory anemia, and various cancer-related diseases [12]. It has been shown that CEP has a half-life of 31.3–36.9 h and reaches steady-state levels after five to six repeated doses of 100 mg/d, with a maximum of 9.6% residual drug after oral administration [13]. The time to reach maximum serum concentrations after a single oral dose of 10–60 mg in healthy adult males ranges from 1.1–2.5 h. After absorption in the body, CEP is extensively metabolized in the liver and distributed to various tissues [10]. Recently, it has been reported that CEP inhibits NF-κB activation [11], lipid peroxidation [14], nitric oxide production, cytokine production and cyclooxygenase expression [15], which are crucial for the inflammatory response. As an “old” drug, CEP has also been reported to be useful in the treatment of COVID-19 [16], with no reported safety concerns or side effects. CEP is soluble in acidic aqueous solutions and some organic solvents such as methanol, ethanol, and DMSO [12]. However, the low water solubility of CEP makes it difficult to be absorbed orally, leading to low bioavailability and reduced efficacy [17]. Therefore, pulmonary administration is a promising therapeutic route.

Pulmonary drug delivery system (PDDS) is a drug delivery system that utilizes aerosol or dry powder particles to deliver drugs directly to the lungs via a specialized drug delivery device, resulting in local or systemic therapeutic effects [18]. Compared with other drug delivery methods, pulmonary drug delivery has the following advantages: (1) For lung diseases, pulmonary drug delivery can help deliver drugs directly to the site of action, resulting in decreased drug doses and distribution in other tissues, and reduced adverse reactions [19]; (2) the lungs contain approximately 300–400 million alveoli with a the total area of 70–100 m^2^, thus providing a large absorption area with abundant capillaries and high blood flow. This enables drugs to rapidly enter the circulation and achieve systemic therapy; (3) the chemical and enzymatic degradation reactions in the lungs are low and less destructive to the drug [20,21], which help the drug avoid gastrointestinal degradation and the first-pass effect of the liver; and (4) compared with the injectable route, pulmonary administration improves patient compliance, and is more convenient and effective [22]. Compared to liquid pulmonary inhalation formulations, DPI is more stable and does not require refrigeration for storage. DPI is easier to administer, without propellants and has fewer irritating effects. DPI uses the patient’s inspiratory force to allow the drug to be deposited deep in the lungs. The optimal aerodynamic diameter of particles for pulmonary inhalation formulations ranges from 1–5 μm [23,24].

Based on the anti-inflammatory and antiviral effects of CEP, and the characteristics of the pulmonary delivery form that make it beneficial for the topical treatment of acute lung injury, we prepared CEP in a pulmonary inhalation delivery form for the treatment of acute lung injury. The major content of the study involved the preparation and characterization of CEP DPIs for pulmonary administration, the successful establishment of an ALI rat model by intratracheal spraying of hydrochloric acid, and the evaluation of the pharmacological effects of the CEP DPIs on ALI via tracheal spraying. We provide a new method and ideas for efficient ALI treatment via pulmonary inhalation (Figure 2).

## 2. Results

### 2.1. Optimized Prescription of Cepharanthine Dry Powder Inhalers

After reviewing the physicochemical properties of CEP, we examined the morphology and the particle size of the CEP API by scanning electron microscopy (SEM) and laser particle size measurement to investigate whether a pulmonary inhalation formulation could be directly prepared from the API. The particle size of the API was around 60 μm (Figure 3A), which did not meet the requirements of the pulmonary inhalation formulation. Therefore, we further ground the CEP API for 30 min using a mortar and pestle by avoiding light to verify whether the particle size could be reduced using the physical method. The particle size decreased to about 12 μm after grinding for 30 min (Figure 3B), but it still did not meet the standard of lung inhalation formulation particle size, which is less than 5 μm.

Neither of the products met the standards for pulmonary inhalation preparations, so we conducted excipient screening and added lactose and mannitol as lyophilized protectants and freeze-dried them together with CEP to identify whether the lyophilized samples met the standards for pulmonary inhalation preparations. Nine groups of prescriptions were designed (Table 1), including lyophilized CEP acetate solution alone; CEP to mannitol mass ratio: 1:1, 1:2, 1:3, 1:5; and CEP to lactose mass ratio: 1:1, 1:2, 1:3, 1:5. After freeze-drying the above prescription for 35 h, the powdered aerosol was obtained. We observed the appearance of the above samples by SEM, and the product obtained after lyophilization of CEP acetate solution was seen to have a loose reticulation structure (Figure 3C). However, the addition of lactose as a lyophilization protectant caused the disappearance of this reticulation structure, and the product adhered to the Petri dish in the form of small particles, and its particle size exceeded 100 μm, which was much larger than that of the CEP API (Figure 3H); hence it was not suitable for making inhalation formulations. Furthermore, by using mannitol as a lyophilized protective agent, a loose textured powdered product with a particle size of about 10 μm was also obtained (Figure 3D–G).

To assess the particle size of the product more accurately, the particle size distribution was measured using two scanning modes of the laser particle sizer (Table 2). Compared to the addition of mannitol as a lyophilization protectant, the CEP acetate solution alone showed the smallest particle size.

After comprehensively examining the appearance morphology and particle size of the above prescription groups, we eventually selected the lyophilized CEP acetate solution alone group as the optimal formulation for the subsequent investigation of lung deposition rate and pharmacodynamic verification.

### 2.2. Fine Particle Fractions and Powderological Investigation of CEP

We investigated the powderological properties of the cepharanthine dry powder inhaler by measuring the bulk density and vibrational density of the powder (Table 3) and calculating its aerodynamic diameter (Da) according to the formula. The aerodynamic diameter is the most important factor affecting the deposition site of drugs, and it is generally considered that particles with Da within 1–5 μm can reach the deep lung [25]. Based on the Chinese Pharmacopoeia 2020 version of the “Inhalation preparation of fine particles, aerodynamic characteristics of the method” (General 0951), “Device 3” (Next generation cascade impactor, NGI) was used to examine the fine particle fraction (FPF) of this product to simulate the distribution of CEP dry powder inhaler in the lungs in vitro (Figure 4). The aerodynamic particle size of the CEP dry powder inhaler was 3.202 μm, and its lung deposition rate was as high as 30.26%, which makes it suitable for lung inhalation.

### 2.3. A Cepharanthine Dry Powder Inhaler Has the Effect of Reducing ALI

Dexamethasone, with its anti-inflammatory, anti-allergic, and immunosuppressive effects, is the most commonly used clinical glucocorticoid and is often used to inhibit the early inflammatory amplification response to ALI [26]. Therefore, we chose dexamethasone as a positive control. In this experiment, we anesthetized rats and induced an ALI model via intratracheal injection of hydrochloric acid [27]. We administered CEP and dexamethasone after 1 h of modeling. Changes in lung appearance and lung histopathology were the most direct treatment observations. In order to study the effect of CEP on the lung tissue of the hydrochloric acid-induced ALI rat model, we removed the lungs after 4 h of administration, observed the damage and took pictures. The results indicated that the lung tissues of the rats in the normal group were pink in color, soft in texture and without bruising; in the model group, the lungs of the rats were dark red, with obvious bruising and slight edema visible; however, no obvious damage was observed in the CEP DPIs, CEP gavage, and DXM groups (Figure 5). Microscopic examination of pathological sections revealed (Figure 5) that the alveolar structure of rats in the normal group was clear and intact, and no obvious unusual changes were observed. In the model group, the alveolar structure of rats was severely damaged, and the alveolar wall was thickened and infiltrated by a large number of inflammatory cells, indicating successful modeling. In contrast, the lung tissue lesions of the rats in each treatment group were significantly reduced compared with the model group, and the inflammatory cell infiltration was also significantly reduced.

### 2.4. Cepharanthine Dry Powder Inhaler Has the Effect of Reducing the Inflammatory Response

Pro-inflammatory cytokines, including tumor necrosis factor-alpha (TNF-α) and interleukin-6 (IL-6), are key inflammatory regulators that cause several lung pathologies and diseases [28]. These pro-inflammatory cytokines can bind to pattern recognition receptors (e.g., Toll-like receptors) on lung epithelial cells and alveolar macrophages as cell damage-associated endogenous molecules (also known as danger-associated molecular patterns, DAMPs) to induce lung inflammation and ALI [29]. When ALI occurs, IL-6 promotes neutrophil recruitment, infiltration, which in turn can mediate lung tissue injury and edema. Whereas TNF-α promotes oxidative stress and nitrosative stress by recruiting inflammatory cells, stimulating the production of inflammatory mediators, inducing airway hyperresponsiveness, and participating in lung inflammation [30,31,32,33,34]. Total protein content is a direct marker of inflammatory response. In this study, we induced ALI in rats by administering hydrochloric acid via intratracheal injection. We found that the levels of IL-6, TNF-α, and total protein in the lung tissues of the model group were significantly higher than those of the normal group (*p* < 0.01); the levels of IL-6, TNF-α, and total protein in the lung tissues of the CEP DPIs group were lower than those of the model group and significantly different (*p* < 0.01); compared with the model group, the levels of IL-6 and total protein in the CEP gavage group were significantly different (*p* < 0.01). Compared with the model group, IL-6 and total protein contents in the CEP gavage group were significantly different (*p* < 0.01), but TNF-α was not significantly different (*p* > 0.05); while the DXM group exhibited significantly reduced IL-6, TNF-α, and total protein levels in the lung tissue of ALI model rats (*p* < 0.01), which was comparable to the CEP DPIs group (Figure 6). Therefore, the inhibition of pro-inflammatory factors IL-6 and TNF-α can potentially alleviate lung tissue damage in rats with ALI. Furthermore, it may also reduce the total protein content in lung tissues, which is presumed to have a protective effect on alveolar epithelial cells. In particular, the effect of the CEP dry powder inhaler was more pronounced, indicating that the powder inhaler form could enhance its utilization within the lungs.

## 3. Discussion

The outbreak of the new coronavirus epidemic has caused a significant global impact. Coronavirus invades by binding to ACE2 receptors in alveolar type II cells, leading to pathological changes in alveolar epithelial cells, and immune hyperactivation triggers a storm of inflammatory factors that ultimately leads to the development of ALI/ARDS [35]. At present, there is no specific treatment for ALI/ARDS. The main clinical treatment is mechanical ventilation, but the mechanical stress during treatment can lead to ventilator-related events and even pulmonary fibrosis, which can significantly increase morbidity and mortality [36]. Other restrictive fluid management systems can significantly improve oxygenation and shorten the duration of mechanical ventilation, but mortality remains high [37]. Therefore, although these supportive treatments can improve ALI/ARDS symptoms, they also cause complications and sequelae and cannot achieve a radical cure. In terms of pharmacological interventions, promising outcomes have been achieved in the latter stages of phase II and phase III clinical trials with the use of β-adrenergic agonists, statins, heparin, or aspirin for the treatment of ALI [38,39]. However, due to their potential adverse effects, these drugs have not yet received approval from the FDA for the treatment of ALI [4].

CEP, being a natural alkaloid, has a favorable safety profile and possesses unique pharmacological effects such as anti-inflammatory, antioxidant, immunomodulatory, anti-parasitic and antiviral effects [11]. It has been suggested that the antioxidant and anti-inflammatory effects of CEP can significantly reduce acute lung injury induced by bilateral lower limb ischaemia-reperfusion (I/R) in rats [40]. Other researchers have also used an erythrocyte anchoring strategy where they encapsulated CEP in chitosan-coated nanoparticles and bound them to erythrocytes in a non-covalent manner. This strategy has demonstrated efficacy in mouse models of ALI, where the nanoparticles were successfully targeted to the pulmonary system, mitigating the inflammatory damage within the lungs [41]. However, most of the current studies on the treatment of ALI with CEP have been in the form of injectable administration, including intraperitoneal and intravenous injections [41,42]. However, CEP has poor oral absorption and low bioavailability due to its hydrophobic nature. A PDDS is a non-invasive technique that can transport drugs directly to the lesion site and is therefore, the most effective route of drug delivery for the treatment of pulmonary diseases. Compared with other pulmonary inhalation forms such as aerosols and nebulized inhalers, powder inhalers are more portable, convenient to use, and the drug exists in a dry powder state which offers better stability [43]. Therefore, in this study, we used the freeze-drying method to prepare CEP as a powder aerosol. Being an alkaloid, CEP has good solubility in acidic solutions. We used acetic acid to adjust the pH of the aqueous solution, which resulted in a pH of 2.75. The solubility of cepharanthine in this solution was good, and after adjustment, the final pH of the solution ranged from 5.0 to 5.5, which made it suitable for use in subsequent animal experiments involving lung administration [44]. Freeze-drying is a common method used to prepare powder aerosol; the vacuum and low-temperature environment make the drug more stable and convenient for long-term preservation [45]. We evaluated the optimal prescription of CEP powder aerosol by using particle size and appearance morphology of the particles as evaluation indexes. Adding mannitol as a lyophilized protective agent yielded a powder that also met the requirements. However, to minimize the use of excipients, the final decision was to lyophilize the CEP acetate solution alone. The aerodynamic particle size of the final aerosol powder ranged from 1 to 5 μm, and the lung deposition rate was 30.26%, which met the requirements of dry powder inhalation to reach the deep lung respiratory tract.

In this research, we found that administration of CEP via both powder and gavage forms could reduce the severity of ALI caused by hydrochloric acid in rats, but the powder aerosol group demonstrated a more significant effect. It was also demonstrated that the anti-inflammatory effect of CEP was mediated by inhibiting the expression of IL-6 and TNF-α. Furthermore, both the powder aerosol and gavage administration of CEP resulted in a decrease in the total protein content in the lung tissue, suggesting a potential protective effect on the alveolar epithelial cells. However, in this study, we used an ALI rat model, and the dosages applied were suitable for rats. There may be different responses between species, and further confirmation of the dose administered is required for further use in large animal models or for application in human subjects. Additionally, for the CEP powder aerosol we developed, there currently is a lack of an appropriate container for long-term storage. In summary, our drug can be sprayed into the rat lung through the trachea; thus, the drug can directly reach the site of injury, which increases its concentration and bioavailability in the lung, thereby improving its therapeutic effect on ALI. Therefore, it is expected to be an effective agent for the treatment of ALI.

## 4. Materials and Methods

### 4.1. Reagents and Materials

We used the following reagents: cepharanthine (purity 98%, Chengdu Jianteng Technology Co., Ltd., Chengdu, China), acetic acid (batch number: 2020825, Sinopharm Chemical Reagent Co., Ltd., Beijing, China), lactose (batch number: 970421, Academy of Military Medical Sciences Pharmaceutical Supply Station, Beijing, China), mannitol (batch number: 20190322, Sinopharm Group Chemical Reagent Co., Ltd., Beijing, China), anhydrous ethanol (batch number: 10009294, Sinopharm Chemical Reagent Co., Ltd., Beijing, China), dexamethasone (purity 99.86%, MedChemexpress, Monmouth Junction, NJ, USA), Isoflurane (batch number: S10010533, Shanghai Yuyan Scientific Instruments Co., Ltd., Shanghai, China), Rat TNF-α ELISA Kit (batch number: ml102828V, Shanghai Enzyme Link Biotechnology Co., Shanghai, China), BCA protein concentration assay kit (batch number: TC 263614, Thermo), methanol (chromatographic purity, Thermo Fisher Scientific Co., Ltd.), and other reagents are analytically pure.

### 4.2. Establishment of a Rat Model of ALI and Drug Administration Protocol

SPF grade standard deviation (SD) male rats, with a body mass of 200 ± 10 g, were purchased from Beijing Keyu Animal Breeding Center. The rats were acclimatized for seven days before the experiment, and the animals were given free access to food and water during the experiment. Animal Certificate of Conformity No.: SCXK (Beijing) 2018-0010. All relevant operations performed in this experiment were carried out under the approval of the Animal Ethics Committee of the Military Medical Research Institute, and the animal management and use were performed in accordance with regulatory requirements.

### 4.3. The Content Determination

In this work, we used the UV spectrophotometric method to analyze the content of CEP. We measured an appropriate amount of cepharanthine standard solution and dissolved it in anhydrous ethanol to create a reserve solution containing about 140 μg/mL of CEP. Of the above reserve solution, 0.1, 0.5, 8, 10, and 12 mL were added to a 25 mL measuring flask, anhydrous ethanol was then added, the solution was shaken well and diluted into standard solutions of 1.4, 2.8, 44.8, 56.0, and 67.3 μg/mL concentrations, and the absorbance was measured at 282 nm. The results were calculated by regression: A = 0.00886 C + 0.18262 (*r* = 0.9988). The linear relationship between concentration and absorbance was observed in the range of 1–70 μg/mL.

### 4.4. Preparation of Cepharanthine Powder Aerosol

To adjust the pH of the aqueous solution and promote salt formation, we added 200 μL of acetic acid to 13 mL of purified water. The solution was then placed in a freeze-dryer at −35 ℃ for pre-freezing for 20 h. We subsequently subjected the solution to a gradient temperature and freeze-drying for a total of 35 h to produce the CEP powder aerosol.

### 4.5. Investigation of the Powderological Properties

#### 4.5.1. Bulk Density and Aerodynamic Particle Size

After lyophilization, the powder was sieved through a 180 mesh, and the geometric particle size and its distribution were determined by a dry laser particle size meter. The sieved powder (about 1 g) was gradually added into a 10 mL measuring cylinder and vibrated 100 times. The weight and volume of the powder were recorded before and after vibrating, and the bulk density and vibrating density of the powder were calculated. The aerodynamic diameter (D_a_) of the powder was then calculated according to the formula.
$$D_{a}=D_{e}\sqrt{\frac{\rho }{\rho 0\;\chi }}$$

D_e_ is the geometric diameter of the powder, ρ is the effective particle density, and the value is 1.26 times the vibrational density; ρ0 is the reference density, equal to 1 g·cm^−3^; χ is the dynamic morphology factor (χ = 1 for spherical shape, in this paper we used 1). The aerodynamic particle size is the most important factor affecting the site of drug deposition, and it is generally believed that particles with Da within 1–5 μm can reach the deep lung.

#### 4.5.2. Scanning Electron Microscopy to Examine the Appearance of Cepharanthine Powder Aerosol

We applied a small amount of test sample on double-sided conductive adhesive and affixed it on a metal plate. The samples were prepared for SEM by electron beam at 40 mA and gold sprayed twice to observe the morphology, size and other indices of the test samples under 5.0 kV voltage.

#### 4.5.3. Measurement of the Lung Deposition Rate

Each capsule contained 20 mg of CEP dry powder inhalation. Ten capsules were then placed in a powder inhaler, connected to a next-generation impact (NGI), and the airflow rate was set at 60 L/min for inhalation testing. Each pellet was pumped twice for 10 s each time. After completion, we rinsed the powder in the device, throat and pre-separator with anhydrous ethanol and collected the solution in three 50 mL volumetric flasks. We rinsed the acceptors of Class 1, Class 2, Class 3, Class 4, Class 5, Class 6, Class 7, and micro-orifice collector in 10 mL, three 25 mL, three 10 mL, and 5 mL volumetric flasks, respectively. The volumetric flasks were filled with anhydrous ethanol to fix the liquid, and the content of each component was determined using the “*Content Determination Methodology*”. The FPF of the dry powder inhalation was calculated according to the following formula [46].
FPF = W_2~7_/W_total_ × 100%

### 4.6. Pharmacodynamic Evaluation

#### 4.6.1. Establishment of a Rat Model of ALI and the Administration Protocol

Twenty-five SPF SD rats were numbered and randomly divided into five groups of five rats each: normal group, model group, CEP dry powder inhaler tracheal administration group, cepharanthine gavage administration group, and positive drug dexamethasone group. After isoflurane anesthesia, we established the ALI model by spraying saline (1.2 mL/kg) into the trachea of rats in the normal group and hydrochloric acid solution (1.2 mL/kg, pH = 1.25) into the trachea of other groups. After 1 h of modeling, model group rats were sprayed intratracheally with CEP powder aerosol 30 mg/kg 3 times to complete the administration; gavage group rats were administered CEP 30 mg/kg; positive control rats were sprayed with dexamethasone 30 mg/kg intratracheally.

#### 4.6.2. Lung Appearance and Histopathological Examination

The appearance and histopathological changes in the lungs were the most direct changes after the treatment. After 4 h of drug administration, we anesthetized the rats with isoflurane and executed them by bleeding the abdominal aorta. We opened the chest, removed the intact lung, observed the damage, and took photographs; we took the upper lobe of the right lung, washed the surface blood and residue with saline (4 °C), dried it with filter paper, and placed it in 10% formaldehyde solution for 24 h. Histopathological morphological changes were observed under a microscope after H.E. staining [47].

#### 4.6.3. ELISA

We took the right middle lung and right lower lung, weighed them, added nine times the amount of saline (4 °C), and grinded the tissues in a high-speed disperser at 10,000 rpm for 20 s. After grinding, we centrifuged the tissues at 5000× *g* for 10 min in a low-temperature high-speed centrifuge at 4 °C, and collected the supernatants in portions and stored them in a refrigerator at −80 °C. The content of TNF-α and IL-6 in the tissue homogenate was determined by the ELISA kit.

#### 4.6.4. Measurement of BCA Content

The BCA protein concentration determination kit was used to determine the protein concentration. Referring to the steps of the assay kit, the protein concentration was determined using the microtiter enzyme standard method, and the samples were diluted eight times to ensure that the sample points fell within 1/2 of the standard curve.

### 4.7. Statistical Analysis

GraphPad Prism 8.3.0 software was used for statistical analysis of the data (one-way ANOVA), and the experimental data were expressed as mean ± SD. *p* < 0.01 indicates a highly significant difference.

## Figures and Tables

**Figure 1 molecules-28-04441-f001:**
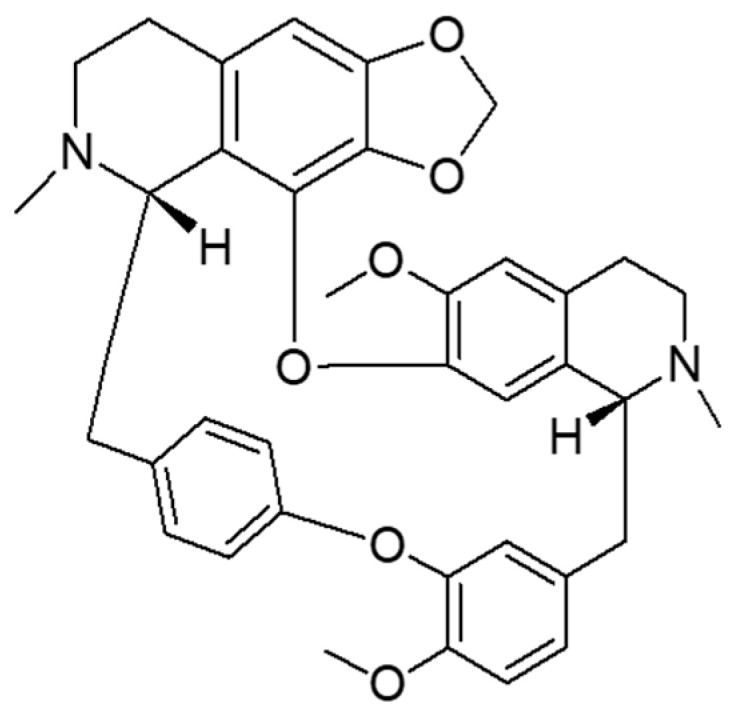
Chemical structure of cepharanthine.

**Figure 2 molecules-28-04441-f002:**
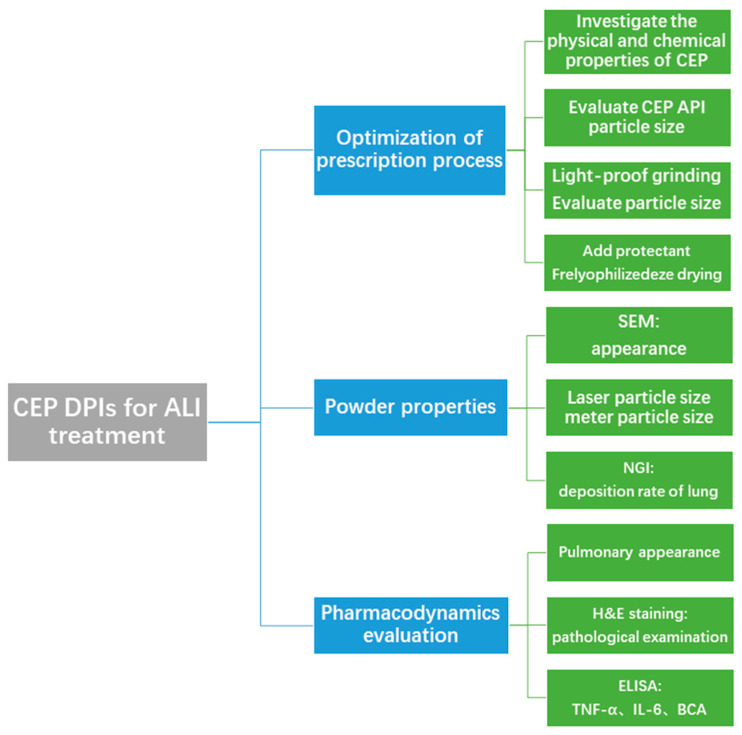
Flow chart of the experimental design.

**Figure 3 molecules-28-04441-f003:**
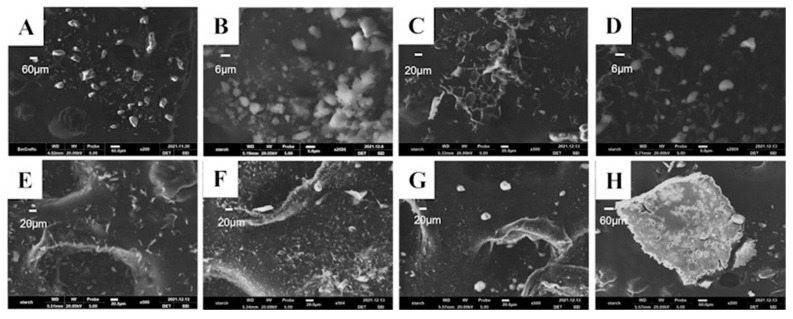
The SEM images of the cepharanthine dry powder inhaler according to different treatment methods. (**A**) Cepharanthine (scale bar = 60 μm); (**B**) Cepharanthine API light-proof grinding (scale bar = 6 μm); (**C**) Cepharanthine acetate solution alone freeze-dried (scale bar = 20 μm); (**D**) Cepharanthine: Mannitol = 1:1 freeze-dried (scale bar = 6 μm); (**E**) Cepharanthine: Mannitol = 1:2 freeze-dried (scale bar = 20 μm); (**F**) Cepharanthine: Mannitol = 1:3 freeze-dried (scale bar = 20 μm); (**G**) Cepharanthine: Mannitol = 1:5 freeze-dried (scale bar = 20 μm); (**H**) Cepharanthine: Lactose = 1:5 freeze-dried (scale bar = 60 μm).

**Figure 4 molecules-28-04441-f004:**
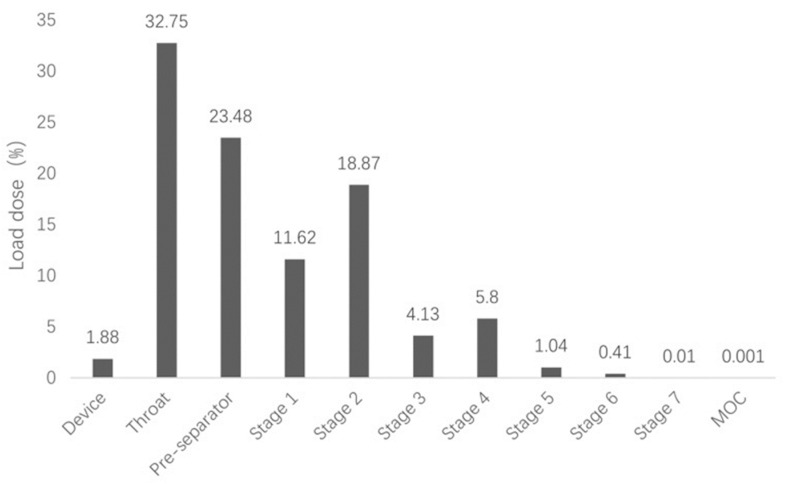
In vitro lung deposition rate of the cepharanthine dry powder inhaler.

**Figure 5 molecules-28-04441-f005:**
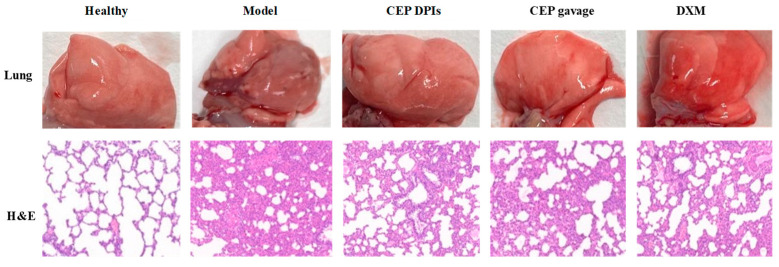
The appearance of lungs visualized by hematoxylin and eosin staining (H & E staining, 100×).

**Figure 6 molecules-28-04441-f006:**
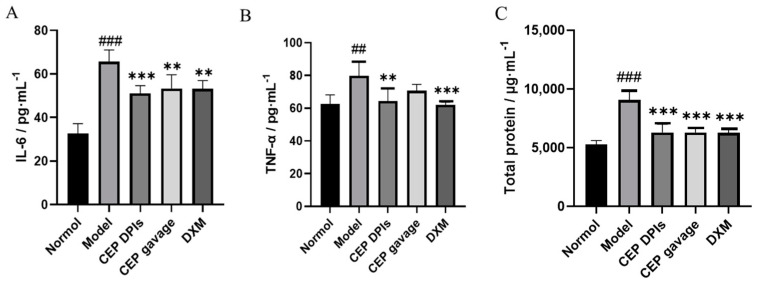
Levels of interleukin-6 (IL-6, (**A**)), tumor necrosis factor-α (TNF-α, (**B**)), and total protein (**C**) in rats’ lungs. *n* = 5, $\bar{x}$ ± s; ^##^
*p* < 0.01, ^###^
*p* < 0.001 vs. normal group; ** *p* < 0.01, *** *p* < 0.001 vs. the model group.

**Table 1 molecules-28-04441-t001:** The major excipient screening of the cepharanthine dry powder inhaler.

Prescription Components	Mass Ratio
Cepharanthine	/
Cepharanthine/Lactose	1:1, 1:2, 1:3, 1:5
Cepharanthine/Mannitol	1:1, 1:2, 1:3, 1:5

**Table 2 molecules-28-04441-t002:** X50 median particle size of mixed freeze-dried samples of cepharanthine and mannitol (*n* = 3, mean ± SD).

Scanning Mode	0.4~87.5 μm	4~875 μm
Cepharanthine	6.27 ± 0.32	7.56 ± 0.37
Cepharanthine/Mannitol = 1:1	7.29 ± 0.58	9.02 ± 0.67
Cepharanthine/Mannitol = 1:2	10.48 ± 1.06	11.19 ± 0.82
Cepharanthine/Mannitol = 1:3	/	7.31 ± 0.72
Cepharanthine/Mannitol = 1:5	/	13.02 ± 0.49

**Table 3 molecules-28-04441-t003:** Characteristics of cepharanthine powders (*n* = 3, mean ± SD).

Parameters	Results
Bulk density (g/cm^3^)	0.142 ± 0.036
Tap density (g/cm^3^)	0.237 ± 0.066
Geometric diameter (μm)	6.27 ± 0.32
MMAD (μm)	3.202 ± 0.092

## Data Availability

The data presented in this study are available on request from the corresponding author.

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
