# Peer review of "Cepharanthine Dry Powder Inhaler for the Treatment of Acute Lung Injury"

_molecules, 2023, doi:10.3390/molecules28114441_

Round 1

Reviewer 1 Report

Reviewer comments

Generally, the articles discuss a good point. The writing style is smooth. The figures are self-explaining. The major organ that is affected by Covid-19 is the lung, even if it is acute effects or after long time. Cepharanthine, is one drug that can increase the effectiveness of other recommended drug for treatment of corona virus plus it has direct effect on inhibiting the entry of the virus into the cell.

·        The author must mention the grinding method in the methodology. What is the technique, is it by mortar or using one of grinding apparatus such as ball mill.

·        How the pH adjusted to pH 5

·        Gavage administration means oral administration, what is the dose and which vehicle is used.

·        Please add the abbreviation (CEP) after cepharanthine in line 42

·        Please replace “cepharanthine” with its abbreviation CEP in lines: 10, 15, 19, 20, 52, 71, 73, 75, 77, 85, 124, 132, 140, 141, 156, 158, 201, 215, 217, 224, 226, 231, 235, 239, 242, 244, 252, 274, 276, 286, 326, 331, 333.

Reviewer 2 Report

  1. Please define any abbreviations when they are first introduced.
  2. Please add references to page 5, lines 135 to 136.
  3. All captions should include n= (sample size) and a scale bar.
  4. Why did you choose to use hydrochloric acid for modeling instead of the more commonly used lipopolysaccharide?
  5. The discussion could be more comprehensive and should be compared with similar studies.

Reviewer 3 Report

The authors have developed dry powder inhaler of a bioactive compound, cepharanthine for the treatment of acute lung injury. Developed powder inhalers were characterized physicochemically and evaluated for pharmacodynamics studies in rats. The proposed work is interesting for the publication in Molecules. However, the manuscript needs major revisions which are suggested below:

Abstract: The quantitative results are missing in the abstract. Kindly include some quantitative results to enhance the readability of the manuscript.

Introduction: authors are advised to include the physicochemical and pharmacokinetic information of studied compound in Introduction. Also include the reported drug delivery systems of studied compound and include the advantages of dry powder inhalers compared to reported drug delivery systems.

Discussion: This section is poor. Kindly explain your results in details and make the proper literature comparison. Please mention clearly how your work is important in comparison to already been reported.

Authors are advised to include the main limitation of work at the end of results and discussion section and just before the conclusion.

Avoid abbreviations before giving their explanation.

Drug release studies: Before performing pharmacodynamics studies in rats, the drug release studies must be performed. Therefore, the authors are advised to perform drug release studies of dry powder inhalers.

Conclusion: The conclusion should be concise and to the point indicating the application of the work.

The language is fine and small grammatical corrections are required.

Round 2

Reviewer 1 Report

·        The authors reply to all comments. No further comments.

Reviewer 2 Report

All the comments are well addressed.

Reviewer 3 Report

The authors have addressed the previous concerns. The revised manuscript is suitable for publication in its present form.

Minor English corrections are required.